# Stopping the Drop: Examining the Impact of a Pilot Physical Literacy-Based Intervention Program on Physical Activity Behaviours and Fitness during the Transition into University

**DOI:** 10.3390/ijerph17165832

**Published:** 2020-08-12

**Authors:** Matthew Y. W. Kwan, Jeffrey D. Graham, Cierra Healey, Natalie Paolucci, Denver M. Brown

**Affiliations:** Department of Family Medicine, David Braley Health Sciences Centre, McMaster University, 100 Main St W, 5th Floor Research, Hamilton, ON L8S 4K1, Canada; grahajd2@mcmaster.ca (J.D.G.); cierrahealey@outlook.com (C.H.); natalie.paolucci@mail.utoronto.ca (N.P.); brownd32@mcmaster.ca (D.M.B.)

**Keywords:** physical literacy, first-year university students, transition, physical activity, fitness, intervention

## Abstract

The move to university is a major life transition associated with precipitous declines in physical activity (PA). While it remains unclear how these declines can be best attenuated, the concept of physical literacy (PL) or enhancements of PL may be a promising modality to promote PA during life transitions. The purpose of the current study was to evaluate the impact of a pilot PL-based intervention on PA and fitness for students transitioning into university. Participants included 65 first-year students (*M*_age_ = 17.85 ± 0.51; *n* = 46 females), enrolled in a quasi-experimental study. Intervention participants (*n* = 26) participated in a 12-week novel movement skills program conducted in a group-based environment. Results from the 2 × 2 repeated measures ANOVA found moderate effects in the time by condition interaction *F*(1,56) = 2.70, *p* = 0.11, η_p_^2^ = 0.08 for PA behaviors as well as for estimated cardiorespiratory fitness *F*(1,61) = 8.35, *p* = 0.005, η_p_^2^ = 0.12. Findings from our pilot program suggest that PL may be an effective modality to help first-year university students maintain fitness and attenuate the declines in PA behaviors when transitioning into university. Similar trials with larger samples are required.

## 1. Introduction

It is well established that there are many positive physical and psychosocial health benefits associated with regular engagement in physical activity (PA) behaviors [1,2]. Participation in PA is related to reductions in the risk of cardiovascular disease, diabetes mellitus, obesity, some forms of cancer, depression, and stress [3,4,5]. Despite the many benefits, approximately 68% of emerging adults are not meeting PA guidelines [6]. The specific transition out of high school has been identified as a period for which significant changes in PA behaviors occur [7,8,9].

The transition from late adolescence into emerging adulthood is often viewed as the first major life transition that an individual will face [10]. For the many that transition into postsecondary education, it will require significant shifts in priorities alongside changing academic and domestic roles and responsibilities [11,12,13]. While many of the health benefits associated with PA tend to be distal in nature, there is evidence to suggest that regular PA has acute health and cognitive benefits to university students [14]; therefore, the specific transition into college or university represents a critical period for which intervention efforts are needed.

The concept of using physical literacy (PL) as a framework to improve PA and health outcomes, including physical fitness, has been a relatively new approach that has garnered a lot of recent attention [15]. It moves beyond an exclusive on motor skills to encompass the competence and confidence to engage in a variety of activities, in a wide variety of environments, and emphasizes the possibility that one can extract positive experiences from movement and learn to value the experience of embodiment. Together, this multidimensional construct of PL is considered necessary to maintain lifelong engagement in PA and physical fitness [16]. Specifically, it posits that an individual with proficient movement skills will feel confident in their participation abilities and will be more motivated to continue to engage in PA behaviors that develop and further strengthen their skills. Knowledge and understanding of the importance of PA as it relates to health and wellbeing will further enhance motivation towards the engagement in physical activities that develop movement skills. Despite its recent popularity, PL has not been well studied empirically in relation to PA behaviors. Interestingly, each of the individual domains of PL (i.e., competence, confidence, motivation, and knowledge) are independently considered to be important correlates of PA during the adolescent and emerging adulthood period (e.g., Self-Efficacy Theory or Self-Determination Theory [17,18,19,20]), and while the concept of PL has seemingly been widely implemented programmatically, particularly for school-aged children in educational and sport contexts, there have been few studies that have applied PL to the late adolescent and emerging adulthood populations.

Taken together, there is good rationale for the development of a PL-based intervention program to target the PA declines typically observed as individuals transition out of high school and into college or university, that is, designing a program that intentionally targets each of the core domains of PL simultaneously: movement competence, confidence, motivation, and knowledge and understanding. It has also been suggested that the environmental contexts in which PL programs are implemented should be considered most salient [21], thus the development of PL can only be effective in settings perceived as fun and enjoyable to individuals. The premise is that individuals will have difficulties in deriving feelings of confidence and the motivation to formulate intentions to engage in physical activities if they are unenjoyable [21]. While each domain should be considered a central focus when designing interventions, PL can further be enhanced through a social environment that provides opportunities for participants to gain confidence from one’s evaluations of others’ perceptions of themselves, to learn vicariously by observing others succeeding in tasks, to create cohesion among the group, and to maintain motivation to continue engaging in the activities over time [22,23,24]. Therefore, by developing a program that effectively targets PL (i.e., salient antecedents to PA) to students transitioning into university, the idea is that it may be able to help attenuate the large declines in PA typically seen.

The purpose of the current study was to develop and evaluate the effectiveness of a pilot PL-based intervention on students’ PA and physical fitness during their transition to their first year of university. It was hypothesized that participants in this program would be able to better sustain their PA behaviors compared to typical students entering first-year university, and to also better maintain their physical fitness. Given that the intervention primarily targets the domains of PL, it was hypothesized that intervention effect on PA and fitness would be mediated through improvements in PL.

## 2. Materials and Methods

### 2.1. Study Design and Participants

The current study utilized a quasi-experimental trial design and included a final sample of 65 first-year university students (*M*_age_ = 17.85 ± 0.51). As a part of the inclusion criteria, students had to be enrolled in their first year of studies, transitioning directly from high school, and residing on campus. Participants in the intervention arm (*n* = 26) were specifically recruited from a designated living learning community called Healthy Active Living (HAL), and participated in the Physical Literacy for University Students (PLUS) intervention program. The HAL community provides student residents with periodic events and seminars throughout the year to provide students exposure to information to promote healthy lifestyle behaviors (e.g., PA, nutrition, sleep), and students within this living learning community were offered the opportunity to take part in the PLUS program on a first-come, first-served basis. To account for selection bias, our control group included participants who were recruited from the HAL community (*n* = 17), as well as from non-HAL (*n* = 22) residences. 

### 2.2. Procedure

Participants were recruited prior to the start of students’ first semester at university. E-mail invitations were first sent out to all incoming HAL students in the summer, with a brief explanation of the PLUS intervention and the study requirements. This included in-person assessments of physical competence and fitness as well as a questionnaire at baseline and follow-up 12 weeks later. Participants that enrolled into the PLUS intervention program signed up to engage in one of two one-hour weekly sessions (i.e., 8:30 am–9:30 am or 9:30 am–10:30 am) during the fall semester. For pragmatic purposes, each intervention session included a maximum of 15 students. Once it reached capacity, all prospective participants were invited to take part as control participants. Additionally, if the PLUS program did not fit into their schedule or if they did not want to commit to the 12-week intervention program, students were asked to participate in the control arm, taking part in baseline and post-intervention testing only. Participants were also recruited throughout welcome week, where research assistants had the opportunity to engage with first-year university students at orientation events at all campus residences to promote participation in the study. 

Students expressing interest in either the PLUS program or as a control participant attended an in-person testing session at the athletic center on campus to complete baseline assessments during the second week of the fall semester. Follow-up testing was completed 12 weeks later, where participants returned for in-person testing. Each assessment took approximately 40 min to complete, and all participants received a $20 gift card for their participation at both baseline and follow-up. This study was conducted in accordance with the Declaration of Helsinki, and all study materials and procedures were approved by the institutional research ethics board. All participants provided written, informed consent prior to their baseline testing session. 

### 2.3. The PLUS Intervention

The PLUS intervention program took place for 12 consecutive weeks, consisting of 60 min sessions once per week that were intentionally designed to introduce novel movement skills through game-based activities. Led by four trained intervention leaders, each session was primarily aimed at enhancing movement competence, motivation, and confidence, as well as knowledge and understanding by engaging in novel movement-based activities. More broadly, the goal was to create a fun and engaging group environment that focused on nontraditional physical activities, and to promote proficiencies in moving on land, air, and water. For example, there were two weeks with aerial activities: one session with team activities on a high ropes course on campus, and another week doing various challenge-based games at a climbing wall. Another session was dedicated to movement in water, including different games with inner-tubes to move within the water. Other weeks’ activities included sessions introducing participants to: “high-intensity interval training (HIIT)” using outdoor equipment on campus; “capture the flag” utilizing the athletic fields; “throwing, catching, and aiming” skill-based games through the popular beach and cottage activities such as KanJam™ Disc Game, Ladderball, and Ring Toss; and “adapted sports” where participants learned and participated in “seated volleyball” and “goal ball” (sports for those with lower limb and visual impairments). A detailed study program manual was developed prior to the start of the intervention. The document specifically outlined the overall goals of the PL-based intervention program, with detailed instructions for each of the activities conducted each week and how these different activities were targeting the movement competence, confidence, motivation, and knowledge and understanding of the participants. The manual also included several contingency plans for alternative activities if there were issues of inclement weather, as well as protocols for dealing with medical emergencies. Following each of the weekly sessions, the intervention leaders also completed an implementation checklist to ensure that the intervention was delivered consistent with the manual outline, and to provide feedback for necessary modifications for future sessions.

## 3. Study Measures

### 3.1. Sociodemographic Factors

Participants were asked to self-report on age, gender, ethnicity, and parental education.

### 3.2. Physical Activity

Self-reported PA was assessed using the International Physical Activity Questionnaire—Short Form (IPAQ-SF [25,26]). Participants were asked to respond to three items and indicate, during the last seven days, on how many days they did: (1) vigorous physical activities; (2) moderate physical activities; and (3) walked for at least 10 min at a time. If they indicated that they participated in an activity for one or more days, they were then prompted to indicate how much time (hours and/or minutes) they usually spent on one of those days participating in that activity. Total time spent in moderate-to-vigorous PA served as the main outcome measure of PA. 

### 3.3. Musculoskeletal Fitness

Musculoskeletal fitness was assessed through standing long jump and grip strength, which are common and valid field-based measures of musculoskeletal fitness [27]. For standing long jump, participants were instructed to jump as far as they could from behind a marked line using a two-foot takeoff and a two-foot landing. Distance was measured to the nearest centimeter from the back of the closest heel to the line. Participants completed three attempts with the longest distance used as the standing long jump performance outcome. Grip strength was measured using a handheld dynamometer (Takei 5401 Hand Grip Digital Dynamometer). To perform the grip strength test, participants stood upright with their arms straight at their sides and the dynamometer was placed in his/her left hand. Participants were instructed to keep their arm straight and squeeze the dynamometer as hard as they could. A research assistant counted to five and recorded the max grip force (in kilograms). The same process was then completed using the right hand and two attempts were made using each hand. The highest force produced in each hand served as the grip strength performance outcome.

### 3.4. Cardiorespiratory Fitness

The Leger 20 m Shuttle Run test was used to represent participants’ cardiorespiratory fitness levels [28,29], which is a valid field-based measure of cardiorespiratory fitness [27]. The test involves running back and forth between two lines set 20 m apart, in synchrony with a sound signal that increases in difficulty (i.e., becomes shorter) over time. The test is terminated when a participant is unable to maintain the set pace for two consecutive sound signals. The number of laps completed and the final stage achieved served as outcomes of aerobic fitness. Cardiorespiratory fitness (CRF) was also predicted using the equation: *y* = 31.025 + 3.238 (maximal speed) − 3.248 (age, years) + 0.1536 (speed × age) [29].

### 3.5. Physical Literacy 

Consistent with Kwan et al. [30], a single measure of PL was calculated using standardized scores (i.e., z-scores) from each domain of PL, which included movement competence, confidence, motivation, and knowledge and understanding (as described below). These scores were then summed, with higher values reflecting greater overall PL. An increase in scores from Time 1 to Time 2 indicates PL increased over time, whereas a decrease in scores indicates PL decreased from Time 1 to Time 2. Internal consistency for the baseline and follow-up composite scores were good (Cronbach’s α’s ranging from 0.77 to 0.79). 

Specifically, movement competence was assessed using the PLAYfun tool [31,32]. The assessment comprises 18 different movement tasks within five domains: (1) running, (2) locomotor, (3) object control—upper body, (4) object control—lower body, and (5) balance, stability, and body control. Confidence, or self-efficacy, for engaging in PA and sports was assessed using two items adhering to recommendations by Bandura [33] for assessing self-efficacy. Motivation was assessed by the Relative Autonomy Index calculated using the Behavior Regulation in Exercise Questionnaire-3 (BREQ-3) [34,35]. Finally, knowledge and understanding were assessed using five items in the BREQ-3 regarding the importance of engaging in regular PA. Internal consistency for the baseline and follow-up scale scores were acceptable (α’s ranging from 0.71 to 0.81).

### 3.6. Data Analysis

All statistical analyses were conducted using SPSS 25. Descriptive statistics were computed for the study variables. Separate one-way analysis of variance (ANOVA) models were computed to assess differences in baseline scores between conditions. Mixed 2 (control vs. intervention) × 2 (Time 1 vs. Time 2) repeated measures ANOVAs were used to examine within-subject changes in the outcome measures (i.e., PA, and musculoskeletal and aerobic fitness,) between study conditions. Significant interactions were decomposed and evaluated using paired-sample *t* tests to examine the change in mean scores within each group. Effect sizes for the repeated measures ANOVA are reported as partial eta squared (*np*^2^) and the values for small, medium, and large are 0.01, 0.06, and 0.14, respectively.

To test the secondary hypotheses regarding whether the change in PL mediates the effect of condition (control/intervention) on the change in PA (depicted in Figure 1) and fitness outcomes (depicted in Figure 2), a series of tests for single indirect (mediation) effects were assessed using Model 4 in the *PROCESS* software macro [36]. Tests for sequential indirect (mediation) effects were assessed using Model 6 in the *PROCESS* software macro [36]. Bias-corrected bootstrap procedures utilizing 10,000 simulations were computed for each mediation analysis [37]. A confidence interval that does not cross zero indicates a significant (*p* < 0.05) indirect (mediation) effect.

## 4. Results

Among the 74 participants that completed baseline assessments, 65 completed the follow-up assessments (88% response rate). Given that there were no significant differences in baseline measures of PA and fitness between responders and nonresponders (all *p*s > 0.05), analyses included only those with both baseline and follow-up assessments (intervention: *n* = 26; control: *n* = 39). The included sample had a mean age of 17.85 ± 0.51 years and was composed of diverse ethnicities; however, participants were predominantly female and entered university from highly educated households. Complete participant characteristics are shown in Table 1. 

### 4.1. Physical Activity Behavior

Overall, the average time spent engaging in PA significantly decreased from 269 min to 241 min per week. Participants in the control condition decreased from an average of 265 min of PA per week to 210 min of PA per week, while those in the intervention had a slight increase in reported PA from 278 min to 290 min of PA per week. Results from the 2 × 2 repeated measures ANOVA showed no main effect for time, *F*(1, 56) = 1.22, *p* = 0.27, η_p_^2^ = 0.02, and while not statistically significant, there was a moderate effect in the time by condition interaction *F*(1, 56) = 2.70, *p* = 0.11, η_p_^2^ = 0.08. Given the exploratory nature of this pilot study, we decomposed the interaction using paired samples *t* tests and found that the change in PA for the intervention group remained relatively stable, *t*(21) = −0.31, *p* = 0.73, whereas a significant decrease for MVPA was observed for the control group, *t*(35) = 2.18, *p* = 0.04.

### 4.2. Physical Fitness

Descriptive statistics summarizing Time 1 and Time 2 musculoskeletal fitness (i.e., standing long jump and grip strength) and aerobic fitness scores (i.e., laps completed, final stage achieved, and estimated CRF) are presented in Table 2. Overall, the results from the 2 × 2 repeated measures ANOVAs revealed no significant main effects for time (*p*s > 0.40), or time by condition interactions (*p*s > 0.20) in changes in standing long jump and grip strength performance. While results of the 2 × 2 repeated measures ANOVAs for the change in aerobic fitness scores revealed no main effects for time (*p*s > 0.05), there were significant time by condition interactions for laps completed, *F*(1, 61) = 10.01, *p* = 0.002, η_p_^2^ = 0.14, final stage achieved, *F*(1, 61) = 8.36, *p* = 0.005, η_p_^2^ = 0.12, and CRF, *F*(1,61) = 8.35, *p* = 0.005, η_p_^2^ = 0.12. Specifically, the number of laps completed increased significantly among participants in the intervention condition, *t*(23) = −2.81, *p* = 0.01, whereas scores decreased among control participants, *t*(38) = 1.35, *p* = 0.19. The final stage achieved also increased significantly among participants in the intervention condition, *t*(23) = −2.81, *p* = 0.01, whereas scores decreased among control participants, *t*(38) = 1.00, *p* = 0.32. Finally, predicted CRF scores increased significantly among participants in the intervention condition, *t*(23) = −2.80, *p* = 0.01, whereas scores remained relatively stable among control participants, *t*(38) = −2.81, *p* = 0.32.

### 4.3. Physical Literacy

Descriptive statistics summarizing Time 1 and Time 2 physical literacy scores are presented in Table 2. While results of the 2 × 2 repeated measures ANOVA showed no main effect for time (*p* = 0.99, η_p_^2^ = 0.00), there was a significant time by condition interaction (*p* = 0.03, η_p_^2^ = 0.08). Although post hoc analyses were in the hypothesized direction, they did not reach statistical significance. For instance, PL scores increased among participants in the intervention condition, *t*(22) = −2.01, *p* = 0.06, and decreased among control participants, *t*(36) = 1.57, *p* = 0.13.

### 4.4. Mediation Analyses

To evaluate our hypothesis that the change in PL mediates the effect of condition (control/intervention) on the change in PA and fitness, separate single mediation analyses were computed (see Figure 1 and Figure 2). Given that there were no clear differences in musculoskeletal fitness, we did not run mediational analyses for these outcomes. Overall, results did not find a significant mediation effect for the changes in PA (95% C.I. = −0.12, 0.77), as well as for laps completed (95% CI = −0.18, 3.16), change in the final stage achieved, (95% CI = −0.05, 0.32), and the change in CRF (95% CI = −0.18, 3.16). Additionally, we ran separate sequential mediation analyses with each aerobic fitness outcome included as the dependent variable, group as the independent variable, and the change in PL (*M*_1_) and the change in PA (*M*_2_) as the mediators (see Figure 3). The results of the sequential mediation analyses again did not yield any significant mediation effects for the change in laps completed (95% CI = −1.26, 0.14), change in the final stage achieved (95% CI = −0.16, 0.01), and the change in CRF (95% CI = −0.46, 0.04). 

## 5. Discussion

Overall results from the current study are promising, suggesting that a PL-based intervention can be an effective strategy for attenuating the typical PA decline observed among students during the transition into first-year university [9], while also helping to maintain students’ aerobic fitness. Specifically, findings suggest that the 12-week PLUS program helped students to maintain a similar level of PA to what they reported prior to entering university, while also showing improvements across aerobic fitness assessment scores. By contrast, students that participated in our control group receiving standard support at university were found to have decreased PA behaviors as well as lower levels of aerobic fitness at the end of the term.

The transition out of high school and into emerging adulthood is accompanied by vast changes for many youths, specifically those that move off to university. Perhaps it should not be surprising that the literature has consistently found this life transition to be associated with major declines in PA behaviors [7,8,38], and that previous research has found first-year university students to encounter new barriers to their participation in PA [12]. The university environment, however, has been often considered to be an ideal setting for intervention efforts [13,39]. Furthermore, there is strong evidence that students often enter university with strong intentions or motivations to be physically active, only to have difficulties translating their positive intentions into behaviors [40]. The development of the PLUS program was intended to help students bridge the intention–behavior gap within the postsecondary setting, using PL as the broader framework for intervention delivery. That is, each session of the program was specifically designed to target novel movement skills, with instructions and support from leaders to facilitate greater confidence, motivation, and knowledge and understanding of the types of activities that are available to students on campus. PLUS, being a weekly organized program designed to expose students to the different activity opportunities on and around campus, likely helped to attenuate the declines in PA upon their entry into university. As was expected, results from the PLUS program found a small-to-moderate effect in the improvement of PL for intervention participants [30]. Contrary to hypothesis, however, we did not find that the intervention effects on PA were mediated by PL. It was likely the case that the current pilot program was underpowered to detect these effects, and more work is needed to investigate how potential mechanisms within the intervention may be impacting participants’ PA behaviors. 

While it is important to understand from a behavioral standpoint the impact of the PLUS program, current findings also highlight a more direct health benefit associated with the intervention. The PLUS participants had significantly greater levels of physical fitness compared to participants in the control group. Specifically, results from the Leger test found significant differences across all indicators of aerobic fitness. A caveat to these findings may be that participants who self-selected into the intervention tended to be somewhat more fit than students in the control condition, and while there was a significant intervention effect on fitness, PL was not a significant mediator. Nonetheless, broader findings do suggest that the intervention had a positive impact not only on PA behaviors but also on aerobic fitness, which is associated with many acute and distal health benefits [41,42]. This is critically important as we know PA tracks well from emerging adulthood into later adulthood [43], and that it becomes increasingly challenging to change behaviors later in life [44]. Future studies with larger samples are needed to replicate current findings and to determine the long-term health benefits associated with PL-based interventions; however, the results of the current study indicate that such an intervention may hold promise in terms of engaging and helping students lead healthy active lifestyles during their transition from high school to university.

While this was the first known study to target PL in university students, there are a number of limitations that need to be considered. First, given the pilot nature of the current study, the sample size was small. This was intentional, as we wanted to determine the feasibility of administering a PL-based intervention under the premise of some space constraints. As a result, the current study is likely underpowered to detect significant differences in reported PA behaviors or a mediating role of PL on fitness. Second, intervention participants were recruited from the HAL community who expressed an interest in health and wellbeing prior to entering university, raising some questions about generalizability to the broader student body. Third, the current study only examined the acute intervention effects, and it would be beneficial for future studies to understand the longer-term impact of the intervention, or if the program were to run over the entire first year of study rather than only the first semester. Fourth, PA was assessed using self-report, which is prone to social desirability biases and recall error. Future studies would benefit from using accelerometers, particularly as they could provide an opportunity to examine how much PA participants engage in during these PL sessions. Finally, we used a convenience sampling frame, including a first-come, first-served approach to recruit participants for the intervention. This likely results in some response biases, as seen in some baseline differences in fitness assessments and in the unequal gender distributions (i.e., 70% female). Therefore, this group may not be representative of the broader student population, and future trials with larger samples and random allocation are required. 

## 6. Conclusions

Overall, the current study found a PL-based intervention program to be effective in helping first-year students attenuate the decline typically observed for PA behaviors and maintain aerobic fitness. Although the underlying mechanisms behind the intervention effects remain unclear, the application of a novel movement skill intervention appears promising in terms of helping to promote healthy active lifestyles for emerging adults as they embark on their transition into postsecondary education. Such programs may be critical to help individuals establish and maintain lifelong engagement in PA as they navigate future major life transitions. 

## Figures and Tables

**Figure 1 ijerph-17-05832-f001:**
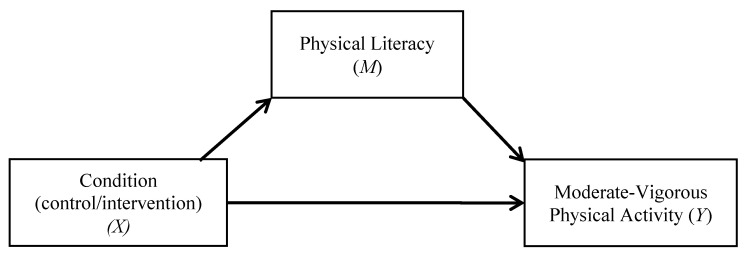
Condition—moderate–vigorous physical activity single mediation model.

**Figure 2 ijerph-17-05832-f002:**
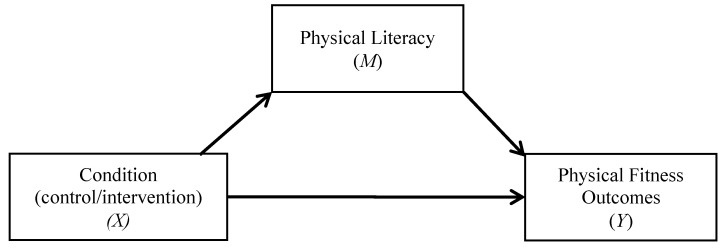
Condition—physical fitness single mediation model.

**Figure 3 ijerph-17-05832-f003:**
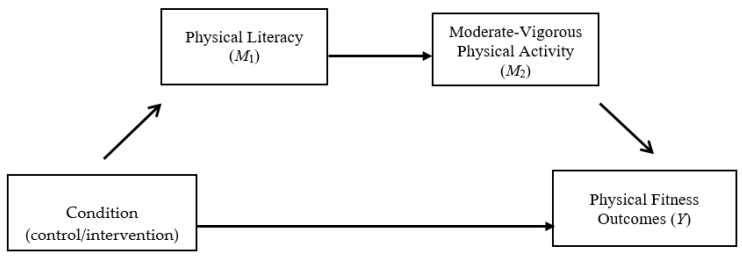
Condition—physical fitness sequential mediation model.

**Table 1 ijerph-17-05832-t001:** Participant Characteristics and Demographics.

Demographic Factor	Total		Control		Intervention		F or X^2^ (*p*-Value)
	(*n* = 65)		(*n* = 39)		(*n* = 26)		
	*n*	%	*n*	%	*n*	%	
Mean Age (SD)	17.85	(0.51)	17.85	(0.49)	17.85	(0.54)	0.00 (0.99)
Gender							0.50 (0.82)
Female	46	70	28	72	18	69	
Male	19	30	11	28	8	31	
Ethnicity							2.59 (0.63)
White/Caucasian	19	29	12	31	7	27	
East Asian	24	37	13	33	11	42	
South Asian	15	23	9	23	6	23	
Black/Latin American	4	6	2	5	2	8	
Other	3	5	3	8	0	0	
Parental Education							3.90 (0.57)
Some Secondary	2	3	2	5	0	0	
Completed Secondary	5	8	2	5	3	12	
Some College	2	3	2	5	0	0	
Completed College	9	14	6	15	3	12	
Some University	4	6	2	5	2	8	
Completed University	43	66	25	64	18	69	

**Table 2 ijerph-17-05832-t002:** Comparison of Physical Activity and Physical Fitness Scores by Condition.

Outcome Measures	Control *M* (SD)		Intervention *M* (SD)		*F*	*p*-Value
Physical Activity						
Time 1	265.25	(108.63)	277.61	(110.19)	0.19	0.67
Time 2	210.23	(152.16)	289.60	(97.49)	5.37	0.02
Interaction					2.68	0.11
Standing Long Jump						
Time 1	169.81	(38.03)	191.77	(29.80)	6.31	0.01
Time 2	172.60	(35.76)	192.69	(36.51)	4.62	0.04
Interaction					0.53	0.47
Grip Strength—Right Hand						
Time 1	32.39	(7.40)	35.68	(7.11)	3.03	0.07
Time 2	32.24	(9.10)	36.05	(7.79)	3.06	0.09
Interaction					0.87	0.35
Grip Strength—Left Hand						
Time 1	30.63	(8.00)	30.76	(9.13)	0.42	0.52
Time 2	31.94	(8.33)	32.84	(8.72)	0.84	0.36
Interaction					1.66	0.20
Leger—Laps Completed						
Time 1	52.53	(18.98)	60.39	(14.92)	3.23	0.08
Time 2	51.28	(18.82)	66.13	(15.89)	10.36	<0.01
Interaction					10.01	<0.01
Leger—Final Stage						
Time 1	6.54	(1.84)	7.31	(1.35)	3.44	0.07
Time 2	6.46	(1.90)	7.92	(1.47)	10.23	<0.01
Interaction					8.36	<0.01
Leger—CRF						
Time 1	40.46	(5.46)	42.75	(3.94)	3.50	0.07
Time 2	40.22	(5.62)	44.59	(4.41)	10.47	<0.01
Physical Literacy						
Time 1	−0.44	(3.50)	0.79	(2.00)	2.53	0.17
Time 2	−0.74	(3.56)	1.22	(1.89)	6.17	0.02
Interaction					4.92	0.03

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
