# Peer review of "Stopping the Drop: Examining the Impact of a Pilot Physical Literacy-Based Intervention Program on Physical Activity Behaviours and Fitness during the Transition into University"

_ijerph, 2020, doi:10.3390/ijerph17165832_

Round 1
Reviewer 1 Report
General comments: this is a novel quasi-experimental study in response to a real-life issue with implications for current and future health of the university student population. While small scale, the study’s results do show promise. My comments are aimed to support the authors (to more effectively communicate this promise to the audience).
Title
Given the exploratory/pilot nature of this research, it is strongly recommended that the title of this manuscript reflects this.
Abstract
Line 11 - Transition used twice in opening sentence.
Be consistent in the use of terminology throughout manuscript (e.g., cardiorespiratory fitness used here but aerobic fitness used elsewhere).
As well as stating the ‘positive’ findings, bring balance to the abstract.
Introduction
Line 43 – change T to t (for therefore). Also, the start of this sentence (lines 41-42) does not currently review to students specifically - please make it clear to which group this research relates.
Given that the current research aimed to evaluate the influence of an intervention upon students’ fitness, there isn’t currently any discussion of the role of fitness in PA and health. Can currently literature be presented to outline associations between fitness, PA participation and health (to ultimately support the measurement of fitness as outcome variables).
Materials and methods
The participant numbers do not appear to total 65, 26 intervention + 43 control [20 HAL and 23 non-HAL] = 69. Please clarify/amend. This has implications for the results (which states that there were 39 in the control group). Did 4 people drop out of the control group? If so, were they from HAL or non-HAL?
Line 194 and 197 – Repeated measures ANOVA
From reading the results, data on gender, ethnicity and parental education were collected. However, in the materials and methods there is not explanation of this. How was this collected? Was this self-reported?
Results
Table 1 – Gender label not in alignment with other text. Also, please remove ‘other’ from Gender classification as not application here. Remove decimal points for Gender, ethnicity and parental education as not applicable (i.e., all whole numbers for N and %).
Table 2 – As with table 1, labels are not aligned consistently.
Discussion
Line 300-302 – can it be concluded that all students in the control group receive standard support at university when some were in HAL?
Line 313 – is ‘school-based environment’ the correct language to use when university is the setting here?
Line 318 – please elaborate in which ways the programme is underpowered.
Conclusion
Line 352 – grammar ‘a novel movement skill intervention’
Author Response
Reviewer 1.
General comments: this is a novel quasi-experimental study in response to a real-life issue with implications for current and future health of the university student population. While small scale, the study’s results do show promise. My comments are aimed to support the authors (to more effectively communicate this promise to the audience).
Response: Thank you for your positive comments and constructive feedback to help us improve the manuscript.
Title: Given the exploratory/pilot nature of this research, it is strongly recommended that the title of this manuscript reflects this.
Response: This is a good point, and we have changed our title to read: “Stopping the Drop: Examining the Impact of a Pilot Physical Literacy-based Intervention Program on Physical Activity Behaviours and Fitness During the Transition into University”
Abstract
Line 11 - Transition used twice in opening sentence.
Response: Thank you for bringing this to our attention. We have subsequently made a change to now read: “The move to university is a major life transition…”
Be consistent in the use of terminology throughout manuscript (e.g., cardiorespiratory fitness used here but aerobic fitness used elsewhere).
Response: We have gone through and changed it to cardiorespiratory fitness throughout.
As well as stating the ‘positive’ findings, bring balance to the abstract.
Response: We have made some modifications to the last section of the abstract to highlight the positive impact. It now reads: “Findings from our pilot program suggest that PL may be an effective modality to help first year university students maintain fitness and attenuate the declines in PA behaviours among students transitioning into university. Similar trials with larger samples are required.”
Introduction
Line 43 – change T to t (for therefore). Also, the start of this sentence (lines 41-42) does not currently review to students specifically - please make it clear to which group this research relates.
Response: We have revised to now read: “While many of the health benefits associated with PA tend to be distal in nature, there is evidence to suggest that regular PA has acute health and cognitive benefits to university students (Bray & Kwan, 2006); therefore, the specific transition into college or university represents”
Given that the current research aimed to evaluate the influence of an intervention upon students’ fitness, there isn’t currently any discussion of the role of fitness in PA and health. Can currently literature be presented to outline associations between fitness, PA participation and health (to ultimately support the measurement of fitness as outcome variables).
Response: This is a good point raised, that had not been specified. We have made changes to the introduction to now read: “The concept of using physical literacy (PL) as a framework to improve PA and health outcomes, including physical fitness, has been a relatively new approach that has garnered a lot of recent attention (Cairney et al., 2019).” The reference has shown below offers a conceptual map of the impact PL can make on PA, Fitness, and other physical and mental health outcomes.
Cairney J, Dudley D, Kwan M, Bulten R, Kriellaars D. Physical literacy, physical activity and health: Toward an evidence-informed conceptual model. Sports Medicine. 2019 Mar 13;49(3):371-83.
Materials and methods
The participant numbers do not appear to total 65, 26 intervention + 43 control [20 HAL and 23 non-HAL] = 69. Please clarify/amend. This has implications for the results (which states that there were 39 in the control group). Did 4 people drop out of the control group? If so, were they from HAL or non-HAL?
Response: Thank you for pointing out the discrepancy. This has now been corrected, as 39 were included in the control condition [17 HAL and 22 non-HAL].
Line 194 and 197 – Repeated measures ANOVA
Response: We have removed the ‘s’ on the end of repeated measures ANOVAs
From reading the results, data on gender, ethnicity and parental education were collected. However, in the materials and methods there is not explanation of this. How was this collected? Was this self-reported?
Response: A subsection describing the self-reported nature of socio-demographic questions have now been added to page 8.
Results
Table 1 – Gender label not in alignment with other text. Also, please remove ‘other’ from Gender classification as not application here. Remove decimal points for Gender, ethnicity and parental education as not applicable (i.e., all whole numbers for N and %).
Response: Thank you. We have made revisions to Table 1, having removed the ‘other’ from gender classification and changed the percentages to be whole numbers only.
Table 2 – As with table 1, labels are not aligned consistently.
Response: We have gone through so that labels are consistently aligned.
Discussion
Line 300-302 – can it be concluded that all students in the control group receive standard support at university when some were in HAL?
Response: It’s a good point raised by the reviewer. We elected to keep this statement in for now, as the HAL community is standard for the university amongst a number of living learning communities that exist on campus. As we specified on page 6, the HAL community consists of only periodic events or seminars occurring through the year, and not a formalized curriculum or program for students. We can change this if the reviewer feels strongly that this statement remains a misrepresentation.
Line 313 – is ‘school-based environment’ the correct language to use when university is the setting here?
Response: We have now revised the sentence to use ‘postsecondary setting’, and now reads: “The development of the PLUS program was intended to help students bridge the intention behaviour-gap within the postsecondary setting, using PL as the broader framework for intervention delivery.”
Line 318 – please elaborate in which ways the programme is underpowered.
Response: We have now added a statement in the limitation section on page 16 that reads: “As a result, the current study will likely underpowered to detect significant differences in reported physical activity behaviours or a mediating role of PL on fitness.”
Conclusion
Line 352 – grammar ‘a novel movement skill intervention’
Response: Thank you, and this has been now corrected.
Reviewer 2 Report
Thank you for the opportunity to review this manuscript. The need to examine physical literacy in relation to physical activity behaviors among the involved age group is well justified. The study addresses an important area of public health and the manuscript has the potential to provide valuable insights.
Comments:
Line 53: Should ‘empirical inquiry’ be ‘physical literacy’?
The recruitment strategy creates challenges with interpreting the success of the intervention. The intervention group was comprised of individuals who already demonstrated an interest in health. The control group were partly comprised of individuals already demonstrating an interest in health + other students without an expressed interest in health. As such, it seems like the experimental groups might be biased in favor of intervention effectiveness (with a greater proportion of health-interested individuals in the intervention group). I think this issue deserves a bit more attention in the discussion - it doesn’t completely devalue the results, but I think the authors can discuss the implications of their recruitment strategy on the findings a bit more, to help readers understand what messages they can more and less confidently take away from this article.
I think the authors need to clarify what the intervention was primarily intending to do: increase physical literacy or increase physical activity. In the introduction, the authors set up the hypothesis that any changes in physical activity that result from baseline to follow-up will be a result of changes in physical literacy, but I’m not convinced the study timeline allows for the testing of that hypothesis. For example, is it possible that the higher physical activity levels reported at week 12 (for the intervention group) reflect that the intervention group took part in activity session 12 during the same week as the follow-up measurements? It’s great to know that taking part in the intervention that involved physical activity sessions resulted in a positive change in weekly physical activity, but I don’t think that finding is addressing the author’s research question, which is focused on the role of physical literacy. It seems likely to me that participating in a physical activity intervention session will increase someone’s physical activity – regardless of any influence on physical literacy. I think it would be helpful for the authors to reflect on these points and revise/add clarifying statements to the article to address them.
Given the author’s focus on physical literacy, I think it’s important for the authors to report baseline and follow-up physical literacy scores, and to discuss whether the intervention was successful at increasing physical literacy.
In the methods section, the authors report that they assessed several psycho-social predictors of physical activity, but don’t report results for those variables. I think it is important to report results for those measured variables.
Author Response
Reviewer 2.
Thank you for the opportunity to review this manuscript. The need to examine physical literacy in relation to physical activity behaviors among the involved age group is well justified. The study addresses an important area of public health and the manuscript has the potential to provide valuable insights.
Response: Thank you for the comment and valuable feedback.
Line 53: Should ‘empirical inquiry’ be ‘physical literacy’?
Response: We made revisions to the sentence to now read “… PL has not been empirically well-studied…”
The recruitment strategy creates challenges with interpreting the success of the intervention. The intervention group was comprised of individuals who already demonstrated an interest in health. The control group were partly comprised of individuals already demonstrating an interest in health + other students without an expressed interest in health. As such, it seems like the experimental groups might be biased in favor of intervention effectiveness (with a greater proportion of health-interested individuals in the intervention group). I think this issue deserves a bit more attention in the discussion - it doesn’t completely devalue the results, but I think the authors can discuss the implications of their recruitment strategy on the findings a bit more, to help readers understand what messages they can more and less confidently take away from this article.
Response: This is a good point raised by the reviewer. It was certainly acknowledged in our methods that we intentionally targeted our recruitment to those that would be most interested, but we did not further explain this in the discussion. We have added a statement in the limitation section that reads: “Second, intervention participants were recruited from the HAL community, whom expressed an interest in health and wellbeing prior to entering university, raising some questions about generalizability to the broader student body.”
I think the authors need to clarify what the intervention was primarily intending to do: increase physical literacy or increase physical activity. In the introduction, the authors set up the hypothesis that any changes in physical activity that result from baseline to follow-up will be a result of changes in physical literacy, but I’m not convinced the study timeline allows for the testing of that hypothesis. For example, is it possible that the higher physical activity levels reported at week 12 (for the intervention group) reflect that the intervention group took part in activity session 12 during the same week as the follow-up measurements? It’s great to know that taking part in the intervention that involved physical activity sessions resulted in a positive change in weekly physical activity, but I don’t think that finding is addressing the author’s research question, which is focused on the role of physical literacy. It seems likely to me that participating in a physical activity intervention session will increase someone’s physical activity – regardless of any influence on physical literacy. I think it would be helpful for the authors to reflect on these points and revise/add clarifying statements to the article to address them.
Response: Thank you for bringing this to our attention. In an effort to increase clarity, we have made revisions to page 5. It now reads: “Therefore, by developing a program that effectively targets PL (i.e., salient antecedents to PA) to students transitioning into university, the idea is that it may be able to help attenuate the large declines in PA typically seen.
The purpose of the current study was to develop and evaluate the effectiveness of a pilot PL-based intervention on students’ PA and physical fitness during their transition to their first year of university. It was hypothesized that participants in this program would be able to better sustain their PA behaviours compared to typical students entering first-year university, and to also better maintain their physical fitness. Given that the intervention primarily targets the domains of PL, it was hypothesized that intervention effect on PA and fitness would be mediated through improvements in PL.”
Given the author’s focus on physical literacy, I think it’s important for the authors to report baseline and follow-up physical literacy scores, and to discuss whether the intervention was successful at increasing physical literacy.
Response: We agree with the reviewer on this point, have added in physical literacy scores to Table 2.
In the methods section, the authors report that they assessed several psycho-social predictors of physical activity, but don’t report results for those variables. I think it is important to report results for those measured variables.
Response: These were used to assess physical literacy, and have now added our results for our composite score of PL in Table 2.
Reviewer 3 Report
There are several problems with this article:
- n very small
- the data expressed in the tables does not make sense with the object of study.
- no initial and final test results are specified.
- the intervention program are leisure activities and activities for the disabled people. No activities that promote adherence to physical activity.
- lack of consult on articles related to adherence to Physical Activities
Check these references
- Kohler, A., Kressig, R. W., Schindler, C. y Granacher, U. (2012). Adherence rate in intervention programs for the promotion of physical activity in older adults: a systematic literature review. Praxis, 101(24), 1535-1547. doi: 10.1024/1661- 8157/a001129
Morgan, F., Battersby, A., Weightman, A. L., Searchfield, L., Turley, R., Morgan, H., Jagroo, J. y Ellis, S. (2016). Adherence to exercise referral schemes by participants – what do providers and commissioners need to know? A systematic review of barriers and facilitators. BMC Public Health, 16, 227. doi: 10.1186/s12889-016-2882-7
Smith, R. A. y Biddle, S. J. H. (1999). Attitudes and exercise adherence: Test of the Theories of Reasoned Action and Planned Behaviour. Journal of Sports Sciences, 17(4), 269-281. doi: 10.1080/02640419936599
Trujillo, K. M., Brougham, R. R. y Walsh, D. A. (2004). Age differences in reasons for exercising. Current Psychology, 22, 348. doi: 10.1007/s12144-004-1040-z
Author Response
Thank you for your review of this paper. We understand that the sample was small and that our current findings are derived from a pilot project. Despite these limitations as identified on page 16, it does represent a first step to intervene with first-year university students. The program itself was targeting the domains of PL, thus, the program included a variety of novel movement based activities rather than more traditional physical activities. Given the low rates of activity, and the wide ranging interests in different activities, there isn’t a clear activity that we would have considered to promote adherence. Rather, as the broader idea of physical literacy suggest, the idea was to move in different ways, to build confidence to move more broadly, learn about the different activities and facilities available to them, and to improve motivation to be physically active. We have included some references similar to the above in that the domains of PL are established antecedents of physical activity maintenance, which was explained with the Cairney et al. (2019) concept paper in Sports Medicine. Given the changes we have made, including to Table 2, we hope that we have made sufficient clarifications to the purpose of the study and its contribution to the overall literature.
Reviewer 4 Report
Authors, I commend you on a well-written and much needed study. After reading your work I do not feel that changes regarding the methods and literature based need adjustments. I was very pleased with the thoroughness and believe this paper should be accepted for publication.
I have provided only a few editorial suggestions. The paper is sound otherwise.
Line 23: replace, but with however
Line 124: replace confidence, and with confidence, as well as knowledge
Line 177: to keep his/her instead of their
Line 323: I would suggest removing the word importantly
Author Response
Reviewer 4.
Authors, I commend you on a well-written and much needed study. After reading your work I do not feel that changes regarding the methods and literature based need adjustments. I was very pleased with the thoroughness and believe this paper should be accepted for publication.
Response: Thank you for the kind words, and support of project.
I have provided only a few editorial suggestions. The paper is sound otherwise.
Line 23: replace, but with however
Response: Our apologies as we did not put line numbers in first. We could not find the but early in the document, which may have been changed with some revisions we have since made.
Line 124: replace confidence, and with confidence, as well as knowledge
Response: This is now been changed to read “confidence, as well as knowledge…”
Line 177: to keep his/her instead of their
Response: We have no replaced their to his/her
Line 323: I would suggest removing the word importantly
Response: We have now removed the word importantly.
Round 2
Reviewer 2 Report
- In the methods section, where the authors describe how PL was measured, please can they provide the range of possible scores for PL? I wasn't expecting to see negative PL scores for the control group, and the intervention group appeared to improve PL by ~0.5 points. I'm not sure how good/bad those scores/changes are, so it would be helpful to know the full range of scores that could have been achieved for PL.
- Line 174: you have the word 'constrol'
- Given how central the variable of PL is to this study, I think the authors need to add a written section in the results, in which they describe how PL changed in the intervention and control groups. I think it would be helpful for the authors to help the readers understand how significant/noteworthy the changes in scores are - the score in the control group changed by ~0.3 points, the score in the intervention group changed by ~0.5 points - are those large or small changes in PL?
- I think the authors need to discuss the change in PL in the discussion. I think two points in particular need to be mentioned:
- Firstly, was the intervention successful at increasing PL? If the change in PL is minor, then the authors need to draw attention to the fact that different intervention strategies for increasing PL among college students need to be explored, and that a reason PL didn't mediate the influence of the intervention on outcome variables might be due to lack of change in PL.
- Secondly, to what extent might the smaller drop in PA among intervention participants be attributed to the change in PL? If the change in PL among intervention participants in negligible, then the authors should comment on other factors that might explain the smaller drop in PA among intervention participants (e.g., greater proportion of health-interested individuals than in the control group, motivating factors associated with intervention and researcher contact).
Author Response
Thank you for your thoughtful comments. We have gone through to provide a response to each of these queries.
- In the methods section, where the authors describe how PL was measured, please can they provide the range of possible scores for PL? I wasn't expecting to see negative PL scores for the control group, and the intervention group appeared to improve PL by ~0.5 points. I'm not sure how good/bad those scores/changes are, so it would be helpful to know the full range of scores that could have been achieved for PL.
Our Response: The PL composite score was calculated by summing z-scores from each domain of PL (i.e., movement competence, confidence, motivation, and knowledge and understanding). Therefore, there is no range for the scores, and they are sample dependent. At Time 1, a negative value in the control condition indicates that participants in this condition has lower PL levels than participants in the intervention condition. However, these scores were not significantly different. Scores in the intervention increased from Time 1 to Time 2 indicating PL improve whereas PL scores decreased among participants in the control condition. We have added information to PL section in the methods to make the above clearer. Based on point #3 the below, we have added information to the results section with regards to how PL scores changed from Time 1 to Time 2 and provided the effect sizes (i.e., partial eta-squared) for the main effect and interaction effect.
- Line 174: you have the word 'constrol'
Our Response: We have corrected this error and change to “control”.
- Given how central the variable of PL is to this study, I think the authors need to add a written section in the results, in which they describe how PL changed in the intervention and control groups. I think it would be helpful for the authors to help the readers understand how significant/noteworthy the changes in scores are - the score in the control group changed by ~0.3 points, the score in the intervention group changed by ~0.5 points - are those large or small changes in PL?
Our Response: We have updated the results section to describe how PL changed over time alongside effect sizes.
- I think the authors need to discuss the change in PL in the discussion. I think two points in particular need to be mentioned:
- Firstly, was the intervention successful at increasing PL? If the change in PL is minor, then the authors need to draw attention to the fact that different intervention strategies for increasing PL among cowllege students need to be explored, and that a reason PL didn't mediate the influence of the intervention on outcome variables might be due to lack of change in PL.
- Secondly, to what extent might the smaller drop in PA among intervention participants be attributed to the change in PL? If the change in PL among intervention participants in negligible, then the authors should comment on other factors that might explain the smaller drop in PA among intervention participants (e.g., greater proportion of health-interested individuals than in the control group, motivating factors associated with intervention and researcher contact).
Our Response: Thank you for these comments, as they are important points that provide a more fulsome explanation behind the findings. The program did improve overall PL, but current findings did not show a significant mediational effect. We discuss how some elements of the program may have helped explain the attenuation in PA declines. It now reads on page 14-15: “The development of the PLUS program was intended to help students bridge the intention behaviour-gap within the postsecondary setting, using PL as the broader framework for intervention delivery. That is, each session of the program was specifically designed to target novel movement skills, with instructions and support from leaders to facilitate greater confidence, motivation, and knowledge and understanding of the types of activities that are available to students on campus. PLUS being a weekly organized program designed to expose students to the different activity opportunities on and around campus likely helped to attenuate the declines in PA upon their entry into university. As was expected, results the PLUS program found a small-to-moderate effect in the improvement of PL for intervention participants [30]. Contrary to hypothesis, however, we did not find that the intervention effects on PA were mediated by PL. It was likely the case that the current pilot program was underpowered to detect these effects, and more work is needed to investigate how potential mechanisms within the intervention may be impacting participants’ PA behaviours.”
Reviewer 3 Report
please see the annexed document

Author Response
Reviewer 3
Suggestions for article improvement:
- About self-determination theory, I advise to review these articles and apply the most relevant to the article.
- A Grounded Theory Exploration of Adversity in Higher Education: An African-American College Graduate Perspective
Author: Johnson, Andrew W.
Publication info: Robert Morris University, ProQuest Dissertations Publishing, 2019. 27736097.
- Leveraging For-Cause Physical Activity Events for Physical Activity Promotion: An Investigation Using Self-Determination Theory
Author: Bernhart, John A.
Publication info: University of South Carolina, ProQuest Dissertations Publishing, 2019. 13901679.
- Relationships between Leisure-Time/Work-Time Physical Activity/Sitting Time and Leisure Satisfaction/Physical Activity Motivation
Author: Kiessling , Peter Brian, II
Publication info: Indiana University, ProQuest Dissertations Publishing, 2019. 22584507.
- Persistence in Higher Education: A Phenomenological Study Exploring the Hurdles Faced by First Generation Students and the Strategies Employed to Overcome Their Barriers
Author: Berner, David
Publication info: Northcentral University, ProQuest Dissertations Publishing, 2019. 22583523.
- The Relationship Between Enjoyment and Physical Activity During a Community Based Youth Physical Activity Program
Author: Keye, Shelby A.
Publication info: University of Massachusetts Boston, ProQuest Dissertations Publishing, 2019. 13857627.
- Relations among Healthy Lifestyle Factors and ADHD in College Students: A Self-Determination Theory Approach
Author: Serrano, Judah W.
Publication info: University of Wyoming, ProQuest Dissertations Publishing, 2019. 13428022.
- Learning from Failure: An Action Research Case Study on Developing Growth Mindset through Academic Risk-Taking in an Athletic Training Program
Author: Rabe, Sarah B.
Publication info: Gardner-Webb University, ProQuest Dissertations Publishing, 2018. 10975828.
- A Case Study of Physical Therapist Assistant Student Motivation and Retention at a Midwest Private Career College
Author: Saggers, Sherry Ellis-Sally
Publication info: Northcentral University, ProQuest Dissertations Publishing, 2018.
- Motivation for Physical Activity Among U.S. Adolescents: A Self-Determination Theory Perspective
- Author: Nogg, Kelsey A.
Publication info: San Diego State University, ProQuest Dissertations Publishing, 2018. 10936215.
- Effects of Work Physical Activity Culture and Basic Needs on Physical Activity Outcomes
Author: Thomas, Erica M.
Publication info: Wayne State University, ProQuest Dissertations Publishing, 2018. 10748986.
- A Qualitative Exploration of College Student Retention: Personal Experiences of Millennial Freshmen
Author: Barker, Kristen
Publication info: Capella University, ProQuest Dissertations Publishing, 2017. 10680766.
- Developing and Validating a New Physical Activity Goal Instrument: Can the Reasons to Exercise (RE<sub>X</sub>) Scale Identify Profiles that Enhance Physical Activity Behaviors?
Author: Kercher, Vanessa Marie
Publication info: University of Idaho, ProQuest Dissertations Publishing, 2017. 10253138.
- Student Perceptions of the Effect of High School Online Physical Education Class Participation on Fitness Knowledge and Motivation for Physical Activity: A Qualitative Case Study
Author: DeCarlo, Darren M.
Publication info: Northcentral University, ProQuest Dissertations Publishing, 2016.
- The Impact of Choice: Exercise Motivation and Physical Activity in College Students Enrolled in Fitness for Life
Author: Tracy, Julia Fera
Publication info: West Virginia University, ProQuest Dissertations Publishing, 2015. 3741901.
- Effects of a Before School Physical Activity Program on Physical Activity, Musculoskeletal Fitness, and Cognitive Function
Author: Knight, Noelle A.
Publication info: East Carolina University, ProQuest Dissertations Publishing, 2015. 1590117.
- Intrinsic motivation & well-being of runners: The role of mindfulness and flow in self-determination theory
Author: Drosman, David J.
Publication info: Alliant International University, ProQuest Dissertations Publishing, 2015. 3703636.
- A modeling approach to identity, motivation, and physical activity participation in former college athletes
Author: Reifsteck, Erin J.
Publication info: The University of North Carolina at Greensboro, ProQuest Dissertations Publishing, 2014. 3624224.
- A Self-Presentation Perspective of Motivation and Physical Activity in University Undergraduates
Author: Totten, Daniel
Publication info: Carleton University (Canada), ProQuest Dissertations Publishing, 2013. MS00204.
- A Qualitative Approach Using the Self Determination Theory To Understand Motivation Within the Concept of Physical Literacy
Author: McClelland, Kathryn
Publication info: University of Ottawa (Canada), ProQuest Dissertations Publishing, 2013. MS25288.
- Self Determination Theory: A Study of the Relationship Between Causality Orientation and Exercise Preference
Author: Manning, Heather
Publication info: The George Washington University, ProQuest Dissertations Publishing, 2013. 1538851.
- Understanding Physical Activity Behavior in Inclusive Physical Education
Author: Jin, Jooyeon
Publication info: Oregon State University, ProQuest Dissertations Publishing, 2013. 3532010.
Our Response: Thank you for providing the suggested student dissertations. The domain of motivation certainly aligns with Self-Determination Theory and has been cited: (Teixeira, P.J.; Carraça, E.V.; Markland, D.; Silva, M.N.; Ryan, R.M. Exercise, physical activity, and self-determination theory: A systematic review. Int. J. Behav. Nutr. Phys. Act. 2012, 9, 78.). Further, other recent concept papers related to PL (e.g., Cairney J, Dudley D, Kwan M, Bulten R, Kriellaars D. Physical literacy, physical activity and health: Toward an evidence-informed conceptual model. Sports Medicine. 2019 Mar 13;49(3):371-83.) go into more detail regarding the underlying psychological processes that help to form PL.
- About PLUS program - Describe what the program consists of.
Our Response: We describe the program of under “The PLUS Intervention” subsection contained within the “Materials and Methods” section. This includes on page 7 & 8 that reads: “The PLUS intervention program took place for 12 consecutive weeks, consisting of 60-minute sessions once per week that were intentionally designed to introduce novel movement skills through game-based activities. Led by four trained intervention leaders, each session was primarily aimed at enhancing movement competence, motivation, confidence, as well as knowledge and understanding by engaging in novel movement-based activities. More broadly, the goal was to create a fun and engaging group environment that focused on non-traditional physical activities, and to promote proficiencies in moving on land, air, and water. For example, there were two weeks with aerial activities: one session with team activities on a high ropes course on campus, and another week doing various challenge-based games at a climbing wall. Another session was dedicated to movement in water, including different games with inner-tubes to move within the water. Other weeks activities included sessions introducing participants to: ‘High Intensity Interval Training (HIIT)’ using outdoor equipment on campus; ‘capture the flag’ utilizing the athletic fields; ‘throwing, catching and aiming’ skill-based games through the popular beach and cottage activities such as KanJam™ Disc Game, Ladderball, and Ring Toss; and ‘adapted sports’ where participants learned and participated in ‘seated volleyball’ and ‘goal ball’ (sports for those with lower limb and visual impairments). A detailed study program manual was developed prior to the start of the intervention. The document specifically outlined the overall goals of the PL-based intervention program, with detailed instructions for each of the activities conducted each week and how these different activities were targeting the movement competence, confidence, motivation, and knowledge and understanding of the participants. The manual also included several contingency plans for alternative activities if there were issues of inclement weather, as well as protocols for dealing with medical emergencies. Following each of the weekly sessions, the intervention leaders also completed an implementation checklist to ensure that the intervention was delivered consistent with the manual outline, and to provide feedback for necessary modifications for future sessions.”
- Used test - Part of the untilizado fitness tests are not validated, so any conclusion cannot be considered valid. The results may be incorrect. There are no results on adherence to FA in the tables and figures. It would be the most interesting.
Our Response: We utilized valid measures of aerobic and musculoskeletal fitness. We provided reference to the validity of the musculoskeletal tests in the previous version (reference number 270). However, in addition to the Leger references (reference numbers 28 and 29), we now have included the statement under the cardiorespiratory fitness section “…which is a valid field-based measure of cardiorespiratory fitness” to make this clearer.